# Microglial STAT1-sufficiency is required for resistance to toxoplasmic encephalitis

**Maureen N. Cowan, Michael A. Kovacs, Ish Sethi, Isaac W. Babcock, Katherine Still, Samantha J. Batista, Carleigh A. O'Brien, Jeremy A. Thompson, Lydia A. Sibley, Sydney A. Labuzan, Tajie H. Harris** *

Center for Brain Immunology and Glia, Department of Neuroscience, University of Virginia, Charlottesville, Virginia, United States of America

* tajieharris@virginia.edu

**Data Availability Statement:** All RNA sequencing data files are available from the GEO database (accession numbers GSE146680, GSE204751, and GSE203655. For graphs that visualize compiled

## Abstract

*Toxoplasma gondii* is a ubiquitous intracellular protozoan parasite that establishes a life-long chronic infection largely restricted to the central nervous system (CNS). Constant immune pressure, notably IFN-γ-STAT1 signaling, is required for preventing fatal pathology during *T. gondii* infection. Here, we report that abrogation of STAT1 signaling in microglia, the resident immune cells of the CNS, is sufficient to induce a loss of parasite control in the CNS and susceptibility to toxoplasmic encephalitis during the early stages of chronic infection. Using a microglia-specific genetic labeling and targeting system that discriminates microglia from blood-derived myeloid cells that infiltrate the brain during infection, we find that, contrary to previous *in vitro* reports, microglia do not express inducible nitric-oxide synthase (iNOS) during *T. gondii* infection *in vivo*. Instead, transcriptomic analyses of microglia reveal that STAT1 regulates both (i) a transcriptional shift from homeostatic to "disease-associated microglia" (DAM) phenotype conserved across several neuroinflammatory models, including *T. gondii* infection, and (ii) the expression of anti-parasitic cytosolic molecules that are required for eliminating *T. gondii* in a cell-intrinsic manner. Further, genetic deletion of *Stat1* from microglia during *T. gondii* challenge leads to fatal pathology despite largely equivalent or enhanced immune effector functions displayed by brain-infiltrating immune populations. Finally, we show that microglial STAT1-deficiency results in the overrepresentation of the highly replicative, lytic tachyzoite form of *T. gondii*, relative to its quiescent, semi-dormant bradyzoite form typical of chronic CNS infection. Our data suggest an overall protective role of CNS-resident microglia against *T. gondii* infection, illuminating (i) general mechanisms of CNS-specific immunity to infection (ii) and a clear role for IFN-STAT1 signaling in regulating a microglial activation phenotype observed across diverse neuroinflammatory disease states.

## Author summary

The brain, an immune-privileged organ, can be invaded and colonized by pathogens such as the opportunistic parasite, *Toxoplasma gondii*. How microglia, the resident immune

experimental group means, data points for individual mice are available on Dryad: doi:10.5061/dryad.fttdz08w2. The remaining data are within the manuscript and its Supporting Information files.

**Funding:** This work was funded by National Institutes of Health grants R01NS112516 and R56NS106028 to T.H.H., T32AI007496 to M.N.C., T32AI007496, T32GM007267, and F30AI154740 to M.A.K., T32AI007046 to S.J.B., T32GM008328 to K.M.S., T32AI007496 to C.A.O., T32AI007496 to I.W.B., and by the UVA Double Hoo Award to M.N.C. and I.S. The funders had no role in study design, data collection and analysis, decision to publish, or preparation of the manuscript.

**Competing interests:** The authors have declared that no competing interests exist.

cells of the brain, provide resistance to infection is an active area of investigation. In this study, we used a genetic approach to generate and study mice with microglia that lack STAT1, a critical transcription factor that confers protection against intracellular pathogens in both humans and mice. We find that despite robust activation and recruitment of immune cells from the blood to the brain during infection, STAT1 deficiency in microglia leads to increased brain parasite burden and uniform lethality in mice when challenged with *T. gondii*. Our bioinformatic analyses also indicate that STAT1 in microglia regulates (i) the expression of large families of genes associated with parasite killing and (ii) a microglial activation state that has been classically seen in neurodegeneration. Our findings identify mechanisms by which microglia contribute to parasite control and contribute to a greater understanding of their cellular physiology during neuroinflammation.

## Introduction

*Toxoplasma gondii* is a ubiquitous protozoan parasite that traffics to the brain, where it establishes a chronic, life-long infection in both humans and mice. Infection with *T. gondii* occurs in two phases: (i) an acute infection wherein parasite disseminates widely throughout host tissues and (ii) a subsequent chronic infection largely restricted to the immune-privileged CNS [reviewed in ref. 1]. Protective immunity during both the acute and chronic phases of *T. gondii* infection converge on cellular responses to the pro-inflammatory cytokine, interferon (IFN)-γ, and its downstream signaling through the transcription factor signal transducer and activator of transcription 1 (STAT1) [2,3]. *Stat1*$^{-/-}$ mice phenocopy *Ifng*$^{-/-}$ mice, succumbing to *T. gondii* infection during the acute stage of infection [4]. Similarly, antibody blockade of IFN-γ during chronic *T. gondii* infection leads to animal mortality, illustrating that interferon signaling is critical for immune resistance during both stages of infection [5].

During both health and disease, microglia, the tissue-resident macrophages of the central nervous system (CNS), play diverse roles in maintaining tissue homeostasis and potentiating inflammation [reviewed in ref. 6]. A large pool of research has focused on studying microglia during neurodevelopmental and neurodegenerative models, yet their functional roles during CNS infection *in vivo* have been less explored. Part of this gap in understanding stems from recent findings that yolk sac-derived microglia are ontogenically, transcriptionally, and functionally distinct from the bone-marrow derived macrophage population that infiltrates the immune-privileged brain during neuroinflammatory states, including CNS infection [7–11]. Microglia have further been shown to rapidly de-differentiate and adopt an inflammatory signature when removed from the brain's unique tissue microenvironment, making *in vitro* approaches inadequate for recapitulating microglial physiology *in vivo* [12,13].

Several studies utilizing pharmacological depletion of microglia by targeting CSF1R during viral CNS infections have collectively revealed that microglia provide protection in controlling viral burden and against host mortality [14–19]. However, CSF1R antagonism alone also depletes macrophages systemically, with multiple studies also pointing to impaired maturation and function of myeloid populations in circulation and lymphoid organs [17,20,21]. CSF1R antagonism has also been reported to elevate baseline levels of pro-inflammatory cytokines in the brains of treated mice [22]. These local and systemic effects thus serve as confounding variables for cell type-specific functional interpretations.

Recently, our group utilized a tamoxifen-inducible cre recombinase system commonly used in neuroinflammatory mouse models to label and assay microglia independently of bone marrow-derived cells [11]. These experiments revealed that microglial but not peripheral

macrophage-derived IL-1α is required for brain parasite burden control. We have extended this experimental paradigm to investigate how the microglial response to IFN-γ impacts neuro-immunity by genetically deleting *Stat1* from this cell type. Given that (i) microglia are the primary hematopoietic cell population present in the steady-state brain, and (ii) IFN-γ-STAT1 signaling is essential for anti-parasitic functions across a wide array of cell types, we hypothesized that microglia serve as the brain's first responders in restricting early CNS infection with *T. gondii* via STAT1-mediated signaling.

Here, we report that despite efficient parasite clearance from peripheral tissues during the acute stage of *T. gondii* infection, mice with microglial-specific *Stat1* deletion succumb to severe toxoplasmic encephalitis following uncontrolled parasite replication within the brain. We also find that, despite this severe pathology observed with microglial genetic targeting, the brain-infiltrating immune compartment displays a robust anti-parasitic activation profile. Our studies thus implicate cell-intrinsic roles for microglia in controlling CNS infection with *T. gondii* and highlight an inability for the peripheral immune system to compensate for STAT1 deficiency in the microglial compartment.

## Results

### Microglial activation and recruitment to *T. gondii* foci within the brain

Previous studies have revealed that myeloid cell recruitment from the blood to the *T. gondii*-infected brain are required for preventing fatal toxoplasmic encephalitis [23]. Because microglia and monocyte-derived macrophages are difficult to discriminate via immunohistochemistry, which offers spatial resolution of the infected brain, we generated CX3CR1^CreERT2/+ x ROSA26^Ai6/Ai6 (WT) mice to specifically fluorescently label microglia with ZsGreen and discriminate them from monocyte-derived macrophages in the inflamed brain parenchyma [24] (**Fig 1A**). We opted to use a genetic *in vivo* approach throughout our studies due to multiple studies characterizing rapid microglial de-differentiation and increased activation when removed from the CNS and analyzed in *vitro* [13,25]. Immunohistochemical analyses illustrated that following intraperitoneal challenge and progression to chronic *T. gondii* infection, microglia attain a classic "amoeboid" morphology throughout the brain, typical of activation during neuroinflammatory states (**Fig 1B and 1C**). We further observed disrupted microglial spatial tiling and increased microglial recruitment to foci of parasite growth (**Fig 1D–1F**). Our fluorescent labeling approach allowed for detection and discrimination of resident microglia from infiltrating macrophages during CNS infection by both confocal microscopy and flow cytometry, permitting the interrogation of microglial functional roles during *T. gondii* challenge (**Figs 1B–1F and S1A–S1D**).

In order to identify transcriptional programs activated specifically by microglia during *T. gondii* challenge, we FACS-sorted and performed bulk RNA-sequencing on microglia isolated from naïve and chronically-infected mice using our ZsGreen fluorescent reporter. Differential gene expression analysis revealed an enriched IFN-γ response signature in microglia purified from infected, relative to naïve mice (**Fig 1G and 1H**). We identified 2,889 upregulated genes and 2,983 downregulated genes in microglia that were significantly differentially expressed during *T. gondii* challenge, relative to uninfected controls. Interestingly, we observed that during infection, microglia displayed a transcriptional activation state suggestive of a disease-associated microglia (DAM) phenotype, which is commonly associated with neurodegeneration and typified by the downregulation of microglial homeostatic genes (including *P2ry12*, *P2ry13*, *Hexb*, *Tmem119*, and *Fcrls*) and concomitant upregulation of microglial disease-associated markers (including *Itgax*, *Apoe*, *Axl*, *Clec7a*) [26,27] (**Fig 1G**). This phenotype emerged when we compared our list of genes differentially expressed by wild-type microglia isolated

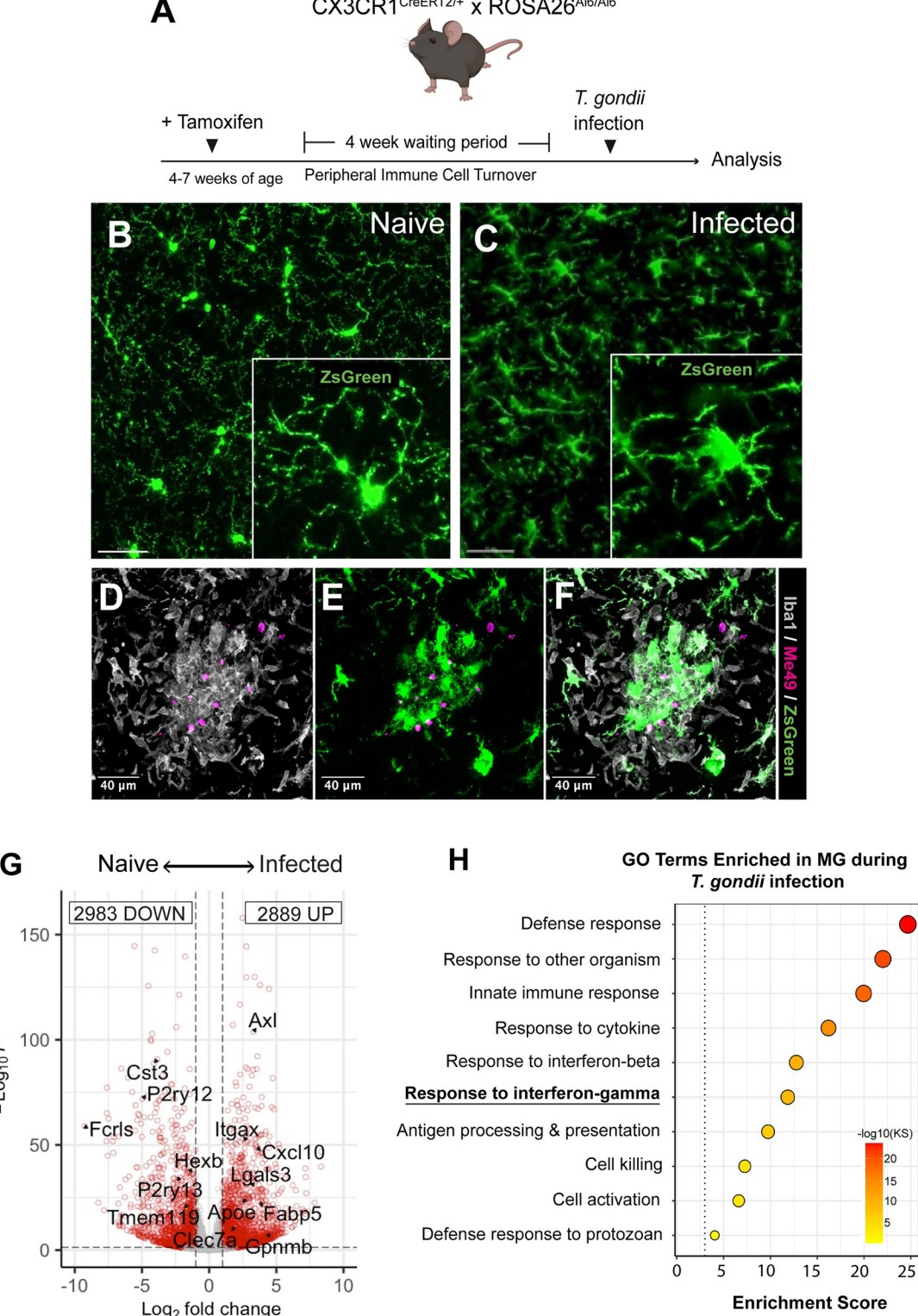

**Fig 1. Microglial activation and recruitment *to T. gondii* foci within the brain.** WT microglia reporter mice were infected with 10 cysts of the Me49 strain of *T. gondii* and were analyzed against naïve controls using IHC and bulk RNA-sequencing. **(A)** Schematic outline for generating microglia reporter mice. **(B, C)** Representative confocal micrographs with zoomed insets of ZsGreen+ microglia (green) in naïve mice **(B)** and at 15 DPI **(C)**, illustrating transition to amoeboid morphology. **(D – F)** Representative confocal micrograph of microglial / macrophage clustering in *T. gondii* foci. Brain sections were immuno-

stained for Iba1 (gray) and Me49 (magenta). **(G)** Volcano plot indicating differentially expressed gene expression and **(H)** gene ontology terms for biological processes statistically over-represented in RNA-sequenced microglia at 28 DPI, relative to naïve state. GO terms were selected based on interest and plotted with enrichment scores that indicate the $-\log_{10}$ of enrichment $p$ value, based on Kolmogorov-Smirnov (KS) analysis. Scale bars indicate 20 μm **(B – C)** or 40 μm **(D-F)**. $n$ = 4–5 mice per group **(G–H)**.

from naïve and infected mice against a core list of common genes that are differentially expressed across multiple neurodegenerative conditions, including mouse models for Alzheimer's Disease, multiple sclerosis, amyotrophic lateral sclerosis, and normal aging [26]. As observed in neurodegeneration models [26], we observed that during *T. gondii* challenge, microglia downregulate homeostatic genes (81% of reported genes) and upregulate DAM genes (71% of reported genes). These data illustrate strong concordance between microglial transcriptional state in *T. gondii* infection and the classical DAM phenotype observed in neurodegeneration (**S2A and S2B Fig**). Given that IFN-γ-STAT1 signaling serves as a dominant pathway driving immune activation and host defense during intracellular parasitic infection [28–31], we focused our efforts on evaluating (i) how microglial *Stat1* deletion impacts host susceptibility to *T. gondii*, (ii) the systems-level effects of this deletion during *T. gondii* challenge, and (iii) a potential role for STAT1 signaling in regulating DAM activation during infection.

## Mice with STAT1-deficient microglia succumb to fatal toxoplasmic encephalitis

Myeloid cells are particularly potent responders to IFN-γ through the production of reactive nitrogen species [32–34] and are capable of serving as antigen-presenting cells during infections [35]. To determine if microglial function during *T. gondii* infection is a similarly essential immune program for defense, we used a *Cx3cr1*-driven tamoxifen (TAM)-inducible system to genetically delete STAT1 from brain-resident microglia, generating MG^STAT1Δ mice (**Fig 2A**).

Excision of STAT1 from microglia in MG^STAT1Δ mice was assessed using multiple approaches, including: (i) ZsGreen reporter expression, (ii) direct quantification of *Stat1* relative mRNA expression, and (iii) quantification of STAT1 functional readouts. On average, 99.5% of CD45^intCD11b+ microglia expressed ZsGreen across TAM-treated animals harboring the microglia reporter construct (**S3A–S3C Fig**). We also observed a 70-85% knockdown in *Stat1* mRNA expression in microglia isolated from naïve tamoxifen-treated (+TAM) compared to corn oil vehicle-treated (No TAM) littermate controls by RT-qPCR (**S3D Fig**). Because STAT1 is a key transcriptional regulator of major histocompatibility proteins [36–38], MHC II expression served as a functional readout of microglial STAT1 excision. We observed that while 98% of WT microglia expressed MHC II at 12 days post-infection (DPI), 18% of microglia from MG^STAT1Δ mice were MHC II+ at this time point, allowing us to use MHC II expression in microglia as a reliable readout for STAT1 excision across experiments (**Figs 2B, S1E and S1F**). In line with previous literature [24,39], these data collectively illustrate that the *Cx3cr1*-driven tamoxifen-inducible system is efficient in targeting brain-resident myeloid cells during CNS infection.

In comparison to WT reporter controls, which display robust resistance to *T. gondii*, we found that MG^STAT1Δ mice succumb to *T. gondii* infection, starting at 17 DPI (**Fig 2C**). MG^STAT1Δ mice also display increased brain parasite burden at 12 and 15 DPI, despite equal brain parasite burden when parasite invades the brain around 8 DPI (**Fig 2D–2F**). Quantitative PCR of *T. gondii* genomic DNA indicated that our observed increase in parasite burden was largely restricted to the brain, as parasite burden in lung, liver, and heart tissue revealed

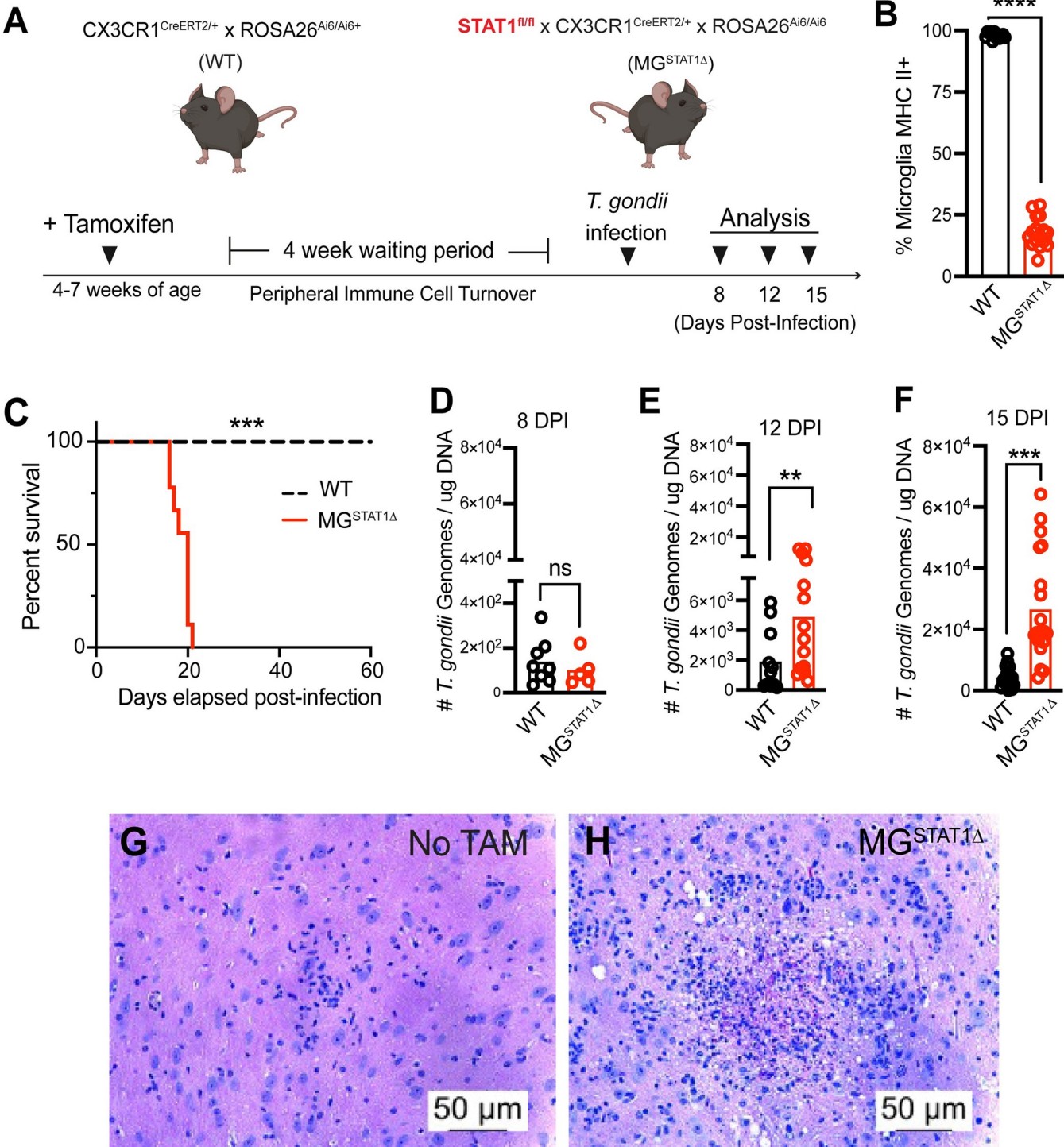

**Fig 2. Mice with STAT1-deficient microglia succumb to fatal toxoplasmic encephalitis. (A-F)** Tamoxifen-treated CX3CR1$^{CreERT2/+}$ x ROSA26$^{Ai6/Ai6}$ (WT) and STAT1$^{fl/fl}$ x CX3CR1$^{CreERT2/+}$ x ROSA26$^{Ai6/Ai6}$ (MG$^{STAT1\Delta}$) mice were intraperitoneally infected with 10 cysts of the Me49 strain of *T. gondii* and analyzed at 8, 12, and 15 DPI. **(A)** Schematic for generating WT and MG$^{STAT1\Delta}$ mice. **(B)** Flow cytometric quantification of microglial MHC II expression at 12 DPI, as a functional readout for STAT1 excision. **(C)** Animal survival curve of WT and MG$^{STAT1\Delta}$ following *T. gondii* challenge. **(D – F)** qPCR analysis of *T. gondii* brain parasite burden quantified by qPCR analysis of homogenized brain tissue at 8- **(D)**, 12- **(E)**, and 15- **(F)** DPI. **(G, H)** Representative brain histopathology observed via H&E staining in litter-mate vehicle **(G)** or tamoxifen-treated MG$^{STAT1\Delta}$ mice at 15 DPI. ns = not significant, $^{**}$ = $p < 0.01$, $^{***}$ = $p < 0.001$, $^{****}$ = $p < 10^{-4}$. Statistical significance was determined by two-way randomized block ANOVA **(B, D-F)** or Kaplan-Meier analysis with $n$ = 12-16 mice per group from 4 experiments **(B)**, $n$ = 6-9 mice per group from 2 experiments **(C)**, $n$ = 5-8 mice per group from 2 experiments **(D)**, $n$ = 11-15 mice per group from 3 experiments **(E)**, and $n$ = 18-20 mice per group from 4 experiments **(F)**. Scale bar = 50 um.

no difference in parasite burden at 8 DPI, around the peak of the acute peripheral infection (**S4A–S4C Fig**). Furthermore, lung, liver, and heart parasite burden decreased substantially from 8 to 15 DPI, indicating robust peripheral clearance of parasite despite increased parasite burden in the CNS (**S4D–S4F Fig**). Histological examination of hematoxylin and eosin-stained brain slices prepared from mice at 15 DPI additionally revealed widespread liquefactive necrosis in MG$^{STAT1\Delta}$ mice treated with tamoxifen (but not vehicle-treated littermate controls), a hallmark of parasite replication-induced tissue destruction (**Fig 2G and 2H**).

Given the severe pathology seen in *T. gondii*-infected MG$^{STAT1\Delta}$ mice relative to controls, we analyzed several additional parameters to examine the specificity of our experimental model. We analyzed ZsGreen expression in circulating immune cells in both WT and MG$^{STAT1\Delta}$ mice by flow cytometry to identify potential off-target STAT1 deletion in brain-infiltrating immune cells. In contrast to >99% ZsGreen labeling efficiency of brain-resident microglia (**S3C Fig**), we observed that ~1% of total circulating immune cells were ZsGreen+ at 12 DPI, and we observed low frequencies of ZsGreen expression across myeloid and T cell populations (**S3E Fig**). No differences were observed in T cell or myeloid cell counts in circulation at this time point (**S5A–S5E Fig**). As an additional parameter for evaluating the CNS tissue-specificity of our model, we analyzed the peritoneal immune response during the acute phase of infection (8 DPI). We observed no changes in the number of myeloid cells, T cells, or their iNOS and interferon-γ production at this time point, indicating that functionally equivalent protective immune responses are able to effectively form at the peripheral site of inoculation (**S5F–S5J Fig**). To assess the potential for differences in microglial activation due to genetic deletion of STAT1 prior to *T. gondii* challenge, we performed morphometric Sholl analysis of microglia in naïve WT and MG$^{STAT1\Delta}$ mice [40]. We found no statistically significant differences in microglial morphology, suggesting comparable baseline microglial activation (**S6A–S6C Fig**). These findings collectively highlight that STAT1-deletion from a single cell type, brain-resident microglia, results in a CNS-specific loss of parasite restriction following infection with *T. gondii*.

## Brain-infiltrating myeloid cells, but not brain-resident microglia, express iNOS during *T. gondii* challenge

Nitric oxide is a potent reactive nitrogen molecular species thought to restrict parasite replication by depleting host cell arginine [41] and targeting parasite proteases [42]. Expression of inducible nitric oxide synthase (iNOS) is the primary mechanism by which cells are capable of producing nitric oxide, and the expression of this synthase is regulated by IFN-γ-STAT1 signaling [43]. Previous studies have implicated microglia in restricting *T. gondii* replication via nitric oxide production [23,44]. Further, iNOS-deficient mice display pathology during chronic CNS but not acute peripheral infection, typified by a loss of parasite restriction and necrotizing lesions throughout the brain – suggesting that microglial-derived nitric oxide serves as a specialized anti-toxoplasmic resistance mechanism specific to brain tissue [45]. However, flow cytometric quantification of iNOS+ cells with the use of our microglia-specific fluorescent reporter indicated that brain-resident microglia fail to express iNOS *in vivo*, regardless of STAT1-sufficiency (**Fig 3A, 3E and 3I**). Instead, we found that > 95% of iNOS expression in the brain is accounted for by the brain-infiltrating myeloid compartment, based on CD45$^{hi}$ CD11b$^{+}$ Ly6C$^{+}$ expression (**Figs 3C and S1G–S1I**). These findings underscore that microglia are not producers of this STAT1-driven mechanism despite a shared tissue microenvironment that triggers potent iNOS expression in other myeloid cells. In addition to the absence of microglial iNOS expression, we quantified and detected no significant changes in

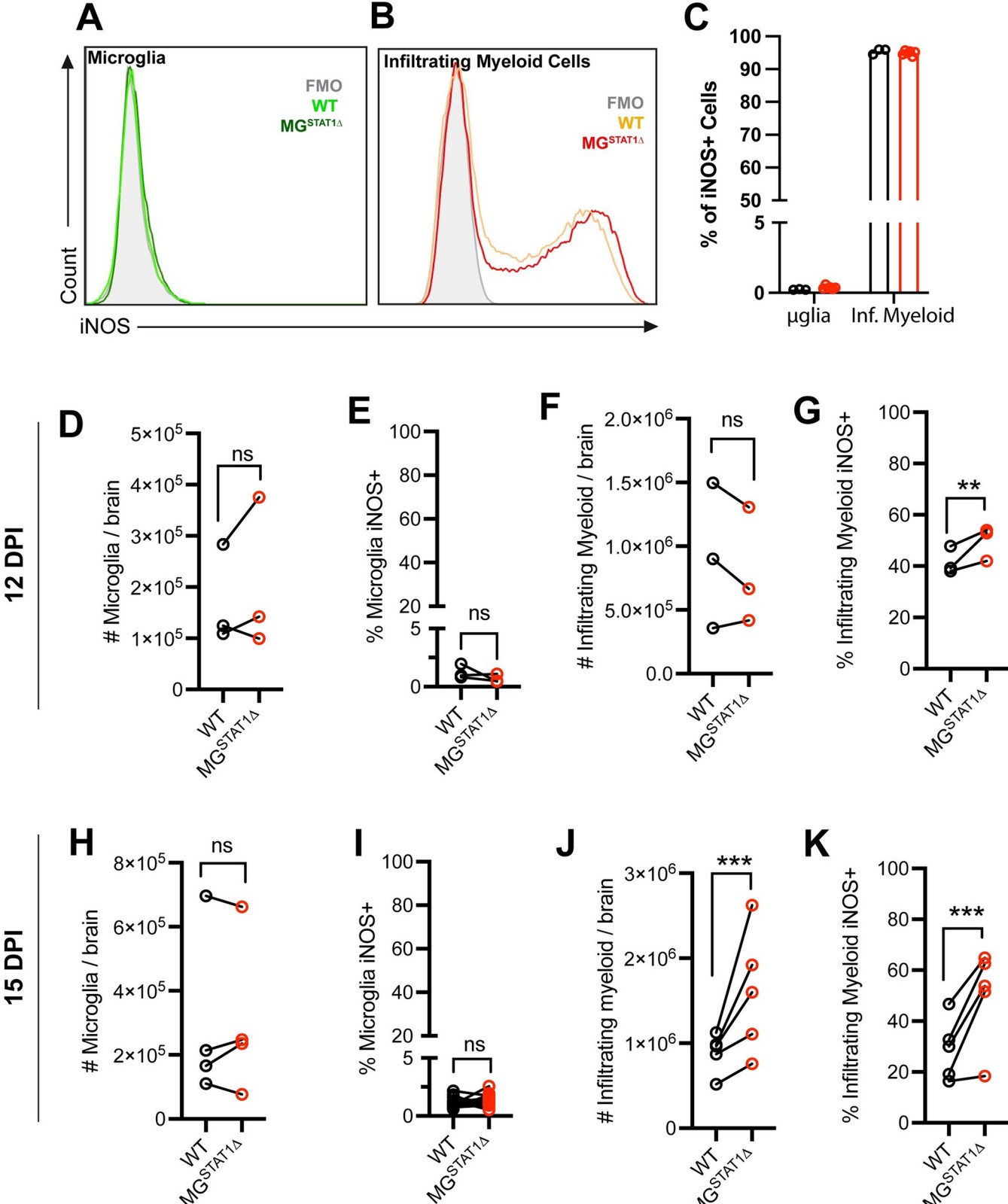

**Fig 3. Brain-infiltrating myeloid cells, but not brain-resident microglia, express iNOS during *T. gondii* challenge.** WT and MG^STAT1Δ mice were intraperitoneally infected with 10 cysts of the Me49 strain of *T. gondii* and brains were analyzed by flow cytometry. Representative histograms indicating iNOS

expression in ZsGreen+ CD11b+ CD45int brain-resident microglia **(A)** and CD11b+ CD45hi Ly6C+ infiltrating myeloid cells **(B)** at 12 DPI. **(C)** Flow cytometric quantification of the total brain iNOS+ cells by myeloid population at 12 DPI. 12 DPI analysis of total number of microglia isolated per brain **(D)**, microglial iNOS frequency **(E)**, number of infiltrating myeloid cells isolated per brain **(F)**, and infiltrating myeloid iNOS frequency **(G)**. 15 DPI quantification of total number of microglia isolated per brain **(H)**, microglial iNOS frequency **(I)**, number of infiltrating myeloid cells isolated per brain **(J)**, and infiltrating myeloid iNOS frequency **(K)**. ns = not significant, ** = $p < 0.01$, *** = $p < 0.001$, **** = $p < 10^{-4}$. Statistical significance was determined by unpaired $t$ test **(C)** or two-way randomized block ANOVA **(D-K)**. $n$ = 3-5 **(C)**, $n$ = 9-10 **(D & F)**, $n$ = 9-13 **(E & G)**, $n$ = 15-16 **(H-I)**, and $n$ = 18-20 **(J-K)** mice per group. **(B-K)** Biological replicates are individual mice, with group means from individual experiments plotted as open circles with black lines connecting experimental and control groups. Data are pooled from 3 experiments **(D-G)**, 4 experiments **(H-I)**, or 5 experiments **(J-K)**. Source data **(D-K)** are provided in a source data file.

microglial number from the brains of WT and MG^STAT1Δ mice at 12 or 15 DPI across several experiments (**Fig 3D and 3H**).

## Microglial STAT1 deletion does not impair the global CNS immune response to *T. gondii*

To examine the potential for microglial STAT1 deletion to impair the development of anti-parasitic effector mechanisms in the brain-infiltrating immune compartment of MG^STAT1Δ mice, we immunophenotyped both infiltrating-myeloid and T cell populations. Previous studies have indicated that both monocyte-derived cells and T cells must traffic into the CNS to prevent fatal toxoplasmic encephalitis [5, 23]. This concept is clearly illustrated by studies revealing that antibody blockade of CD4+ and CD8+ T cells or monocytic CCR2 results in an inability for these protective immune populations to enter the brain and a subsequent loss of parasite control and animal mortality during chronic CNS infection with *T. gondii* [5,23].

**Brain-infiltrating myeloid cells.** At 12 DPI, we observed no difference in the number of brain-infiltrating myeloid cells based on CD45^hi CD11b^+ Ly6C^+ expression, indicating that circulating myeloid cells are able to effectively traffic to the *T. gondii*-infected brain despite microglial STAT1-deletion (**Fig 3F**). We further found a statistically significant increase in the iNOS expression of these myeloid cells, suggestive of increased, rather than decreased, activation and effector functions of the infiltrating myeloid compartment in MG^STAT1Δ mice relative to WT controls (**Fig 3G**). At 15 DPI, there was a clear increase in brain-infiltrating myeloid cell recruitment to the brain, with a further increase in their ability to express the anti-parasitic effector protein, iNOS (**Fig 3J and 3K**).

**Brain-infiltrating CD4+ and CD8+ T cell activation.** In contrast to most peripheral macrophage populations, only a small subset (< 3%) of brain-resident microglia express the professional antigen presentation molecule, MHC II, at baseline [46]. Because STAT1 is a primary transcriptional regulator of MHC II expression [36–38], we investigated MHC II expression on microglia, along with the expression of additional antigen presentation-related molecules (MHC I, CD80, and CD86) that are required for antigen-specific T cell activation [36,37]. In naïve mice, we find that few microglia express any of these antigen presentation molecules (**S7A Fig**). During *T. gondii* infection, however, the expression of MHC I, MHC II, and CD86 was upregulated by microglia in a STAT1-dependent manner (**S7A–S7D Fig**). Despite the STAT1-dependency of these antigen presentation molecules, we quantified no impairment in T cell activation at multiple time-points during infection in MG^STAT1Δ mice. At 12 DPI, MG^STAT1Δ mice displayed no statistically significant changes in CD3^+CD4^+ or CD3^+CD8^+ T cell numbers within the brain, and they showed a two-fold increased frequency of IFN-γ+ expression of both T cell sub-types by flow cytometry, relative to WT controls (**Figs 4A–4D and S1J–S1M**). By 15 DPI, MG^STAT1Δ mice had a three- and two-fold increase in CD3^+CD4^+ and CD3^+CD8^+ T cell numbers in the brain, respectively, with an increased frequency of CD4+ T cell IFN-γ production (**Fig 4E–4H**). To analyze antigen-specific T cell responses, we used a tetramer approach to quantify CD4+ T cells specific for the *T. gondii*

peptide AS15 by flow cytometry [47]. We observed a statistically significant increase in the number of AS15 tetramer CD4+ T cells in MG$^{STAT1\Delta}$ mice, relative to WT controls (**S8A–S8C Fig**). Collectively, analyses of the T cell presence and functional responses in MG$^{STAT1\Delta}$ mice indicate increased, rather than decreased, T cell activation. Further, these results do not suggest that a loss of microglial antigen presentation machinery underpinned the inability to restrict parasite burden in MG$^{STAT1\Delta}$ brains.

**Whole brain RNA analysis.** We performed RT-qPCR analysis of mRNA isolated from homogenized brain tissue to examine the expression of a larger panel of immune mediators of resistance against *T. gondii*. In addition to increased *Nos2* (iNOS) gene expression at the tissue-level, MG$^{STAT1\Delta}$ mice showed an overall equivalent (12 DPI) or increased (15 DPI) expression of several chemokines (*Ccl2*, *Ccl5*, *Cxcl9*, *Cxcl10*), pro-inflammatory cytokines (*Ifng*, *Il6*, *Tnfa*), and adhesion molecules (*Icam1*, *Vcam1*) identified in previous literature as conferring resistance against *T. gondii* infection [5,11,48–53] (**S9A and S9B Fig**). Together, these data indicate that a robust set of anti-parasitic immune mediators are present at the tissue-level, and suggest that neither the brain-infiltrating immune compartment nor other brain-resident cell types are able to compensate for a STAT1 signaling defect within the brain's microglial compartment. We thus focused further efforts on identifying microglial-intrinsic mechanisms of *T. gondii* restriction.

## STAT1-deficient microglia display transcriptionally impaired cell-intrinsic immune activation and effector capacity

In order to attain an unbiased transcriptional overview of how STAT1 regulates microglial immune effector capacity, we FACS-sorted and performed RNA-sequencing on brain-resident microglia from *T. gondii*-infected WT and MG$^{STAT1\Delta}$ mice. (**Fig 5A–5F**). Gene ontology (GO) analysis for biological pathways suggested that relative to WT microglia, STAT1-deficient microglia failed to display the transcriptional signatures of immune defense responses, with GO terms including "response to interferon" and "defense response to protozoan" (**Fig 5A**). Paired differential gene expression analysis indicated that 1,261 genes were significantly enriched in WT microglia, and 831 genes were significantly enriched in STAT1-null microglia (**Fig 5B**). Two key themes emerged from our differential gene expression analysis of microglia isolated from *T. gondii*-infected brains: (i) STAT1-null microglia display an impaired ability to acquire a disease-associated microglia (DAM) phenotype observed across several neuroinflammatory disease states [26,27], and (ii) these cells further fail to express key cytosolic genes required for killing intracellular parasite in a cell-intrinsic manner.

**STAT1-deficient microglia fail to acquire a cell activation phenotype conserved across diverse neuroinflammatory states.** Given our observations that microglia in the *T. gondii*-infected brain display strong transcriptional overlap with DAMs observed during neurodegeneration, we sought to determine whether STAT1 signaling regulated the DAM phenotype (**Figs 1G and S2**). Similarly to our transcriptomic analysis of WT naïve vs. infected microglia, we compared our WT vs MG$^{STAT1\Delta}$ differential gene expression dataset to a dataset of DAM genes shared across several neurodegenerative disease states [26]. Because the DAM phenotype is characterized by both the downregulation of microglial homeostatic genes and the upregulation of unique disease-associated markers during neuroinflammation, we explored both features of the transcriptional signature. We observed during *T. gondii* challenge that relative to wildtype microglia, STAT1-deficient microglia: (i) failed to downregulate 23.5% of all microglial "homeostatic genes" (including *Cx3cr1*, *Tgfb1*, and *Fcrls*) and (ii) failed to upregulate 32% of all DAM-specific genes (including *Itgax*, *Axl*, *Gpnmb*, and *Cybb*) reported in *Krasemann et al. 2017* (**Fig 5C and 5D**). In addition to highlighting that STAT1-deficient microglia retain a more homeostatic and less

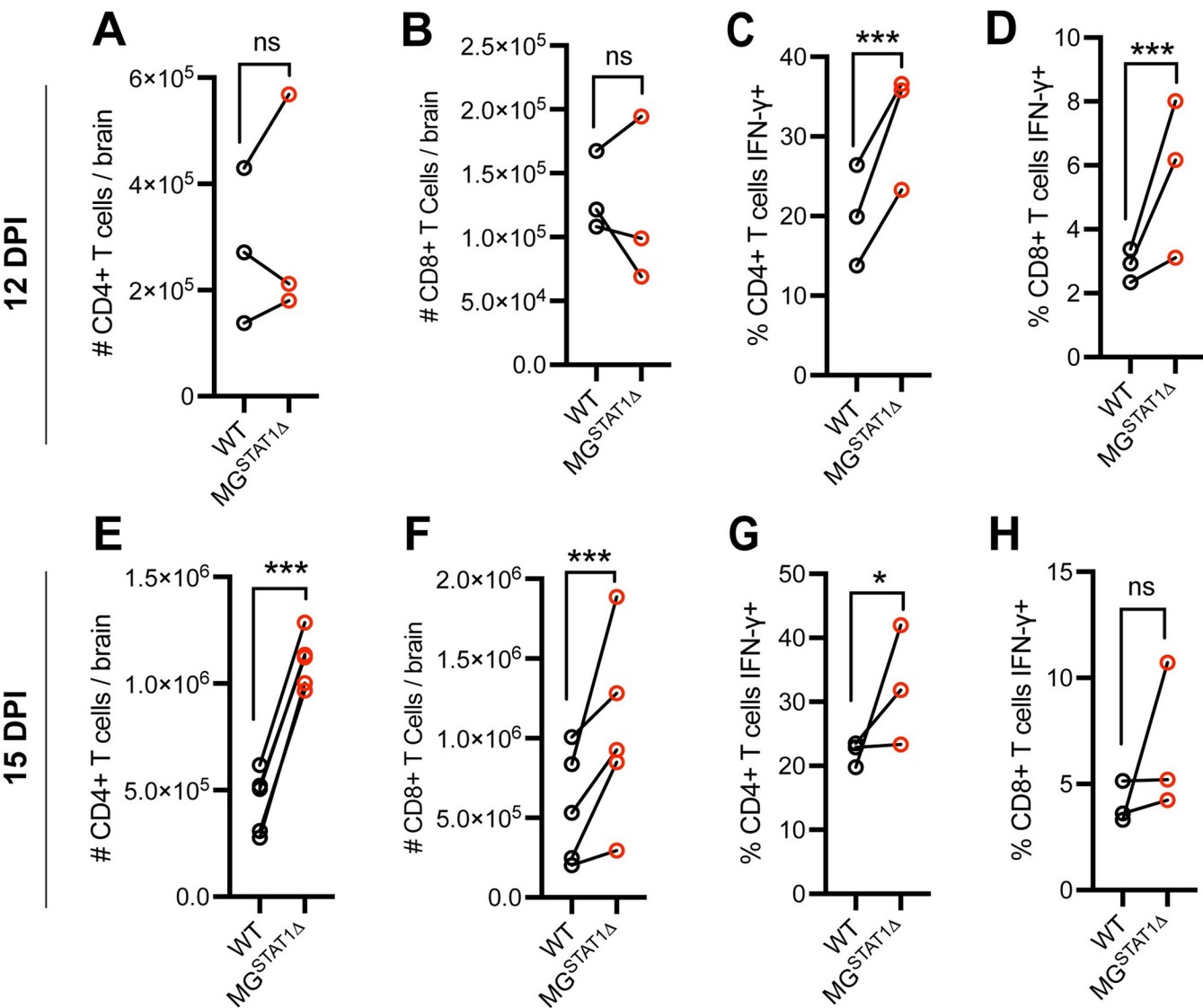

**Fig 4. Brain T cell responses are increased in MG<sup>STAT1Δ</sup> mice relative to WT controls.** WT and MG<sup>STAT1Δ</sup> mice were intraperitoneally infected with 10 cysts of the Me49 strain of *T. gondii* and brains were analyzed by flow cytometry at 12 and 15 DPI. (**A-B**) CD3+CD4+ and CD3+CD8+ cell count enumerated for whole brain and (**C–D**) IFN-γ protein expression at 12 DPI. (**E-F**) CD3+CD4+ and CD3+CD8+ cell count enumerated for whole brain and (**G-H**) IFN-γ protein expression at 15 DPI. ns = not significant, and *** = p < 0.001. Statistical significance was determined by two-way randomized block ANOVA. Data are pooled from 3 (**A-D** & **G-H**), or 4 (**E-F**) separate experiments, with *n* = 9-10 (**A-B**), *n* = 11-13 (**C-D**), *n* = 12-16 (**E-F**), and *n* = 9-12 (**G-H**) mice per group. Biological replicates are individual mice, with group means from individual experiments plotted as open circles with black lines connecting experimental and control groups. Source data (**A-H**) are provided in a source data file.

inflammatory transcriptional signature relative to controls during *T. gondii* infection, these data point to STAT1 signaling as an integral component of a shared microglial transcriptional profile that is conserved across varying models of neuroinflammation.

**STAT1-deficient microglia fail to upregulate genes encoding critical anti-parasitic cytosolic proteins.** To determine specific STAT1-dependent mechanisms that could explain a functional loss of parasite restriction in MG<sup>STAT1Δ</sup> mice, we performed further differential gene expression analysis. We analyzed each of the genes within the Gene Ontology term, "Defense response to protozoan" (GO:0042832) and found that 44% of annotated genes were expressed at significantly lower levels in STAT1-null microglia (**Fig 5E**). Amongst the most highly

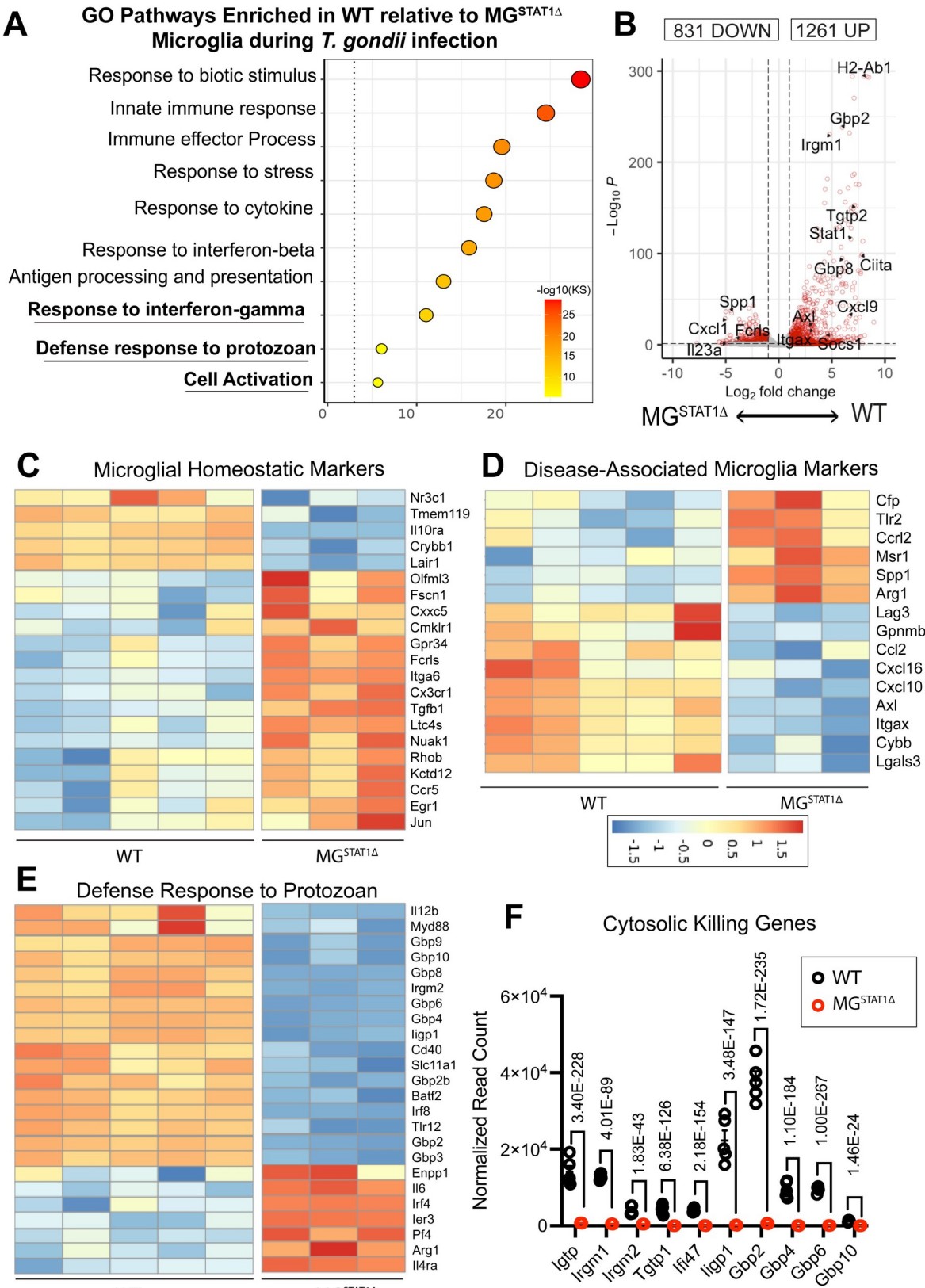

**Fig 5. STAT1-deficient microglia fail to upregulate genes encoding critical anti-parasitic cytosolic proteins.** WT and MG$^{STAT1\Delta}$ mice were intraperitoneally infected with 10 cysts of the Me49 strain of *T. gondii*, and brain-resident microglia were FACS-sorted and RNA-sequenced at 12 DPI. (**A**) Gene ontology (GO) terms statistically enriched in WT relative to MG$^{STAT1\Delta}$ microglia. GO terms were selected based on interest and plotted enrichment scores indicate the -log$_{10}$ of enrichment p value, based on Kolmogorov-Smirnov (KS) analysis. (**B**) Volcano plot indicating differential gene expression between microglia isolated from WT and MG$^{STAT1\Delta}$ mice. (**C-D**) Heat maps displaying VST-normalized, hierarchically-clustered significantly differentially expressed microglial homeostatic genes (**C**) and disease-associated microglia genes (**D**) previously reported as markers of disease-associated microglia (DAM) across neuroinflammatory conditions. (**E**) Heatmap displaying genes from the GO Term "Defense response to protozoan" (GO:0042832) in WT vs. MG$^{STAT1\Delta}$ mice. (**F**) Normalized read count of IRG- and GBP-family proteins expressed by microglia and differentially regulated by STAT1, with *p* values indicating BH-adjusted *p*-values from the full gene expression analysis in DESeq2. *n* = 3-5 mice per group (**A-F**).

differentially expressed genes between WT and MG$^{STAT1\Delta}$ microglia, we observed robust STAT1-dependent expression of several cytosolic anti-parasitic killing genes (**Fig 5E and 5F**). These genes included immunity-related GTPases (IRGs) and guanylate-binding proteins (GBPs), both families of which have been implicated in cooperatively contributing to the mechanical killing and intracellular clearance of *T. gondii* from both hematopoietic and non-hematopoietic cell types [54–58]. We identified several additional IRGs (*Igtp*, *Irgm1*, *Tgtp1*, *Ifi47*) and *Gbp10* that were expressed in microglia in a STAT1-dependent manner (**Fig 5F**). These data identifying the lack of large families of cell-intrinsic parasite killing genes, in tandem with a lack of observable effects of microglial STAT1 deletion on the global, cell-extrinsic immune response, support the hypothesis that cell-intrinsic parasite killing may serve as a primary mechanism of STAT1-dependent microglial parasite restriction. In the absence of this cytosolic restriction, we hypothesized that microglial STAT1-deficiency would generate a replicative niche in which *T. gondii* can expand.

## Microglial STAT1-deficiency results in a skewing of *T. gondii* toward its replicative form

Consistent with the hypothesis that microglial STAT1 deletion provides a replicative niche for the parasite, we observed increased microglial co-localization with large foci of reactive *T. gondii* lesions in the brains of MG$^{STAT1\Delta}$ mice, which has been previously described as characteristic of the highly-replicative and lytic tachyzoite form of *T. gondii*, rather than its semi-dormant bradyzoite (cystic) form [59] (**Fig 6A and 6B**). We then analyzed the prevalence and ratio of these two infectious forms of the parasite in MG$^{STAT1\Delta}$ and WT mice via RT-qPCR. Three primers were designed for *T. gondii*-specific genes: (i) *Sag1*, a gene expressed selectively in tachyzoites (replicative form); (ii) *Bag1*, expressed by slowly replicating bradyzoites (quiescent form); and (iii) *Act1*, expressed by both forms of *T. gondii* [60–62]. While WT brains displayed a one-fold increase in *Bag1* to *Act1* ratio relative to MG$^{STAT1\Delta}$ brains, we observed a nearly 150-fold increase in the tachyzoite gene *Sag1* normalized to *Act1* at 15 DPI in MG$^{STAT1\Delta}$ relative to WT brains, and thus a 300-fold increase in the ratio of *Sag1* relative to *Bag1* with STAT1-deficiency (**Fig 6C–6E**). These data confirm that relative to WT controls, MG$^{STAT1\Delta}$ brains are highly skewed toward a replicative and lytic form of *T. gondii*. Further consistent with microglial STAT1-deletion permitting a parasite replicative niche, we were able to identify microglia filled with tachyzoite rosettes via confocal microscopy in the brains of MG$^{STAT1\Delta}$ but not WT mice (**Fig 6F and 6G**).

## Discussion

Here, we find that abrogation of STAT1 signaling in brain-resident microglia results in a severe susceptibility to CNS infection with *T. gondii*, despite robust immune effector functions in the brain-infiltrating immune compartment and efficient parasite clearance in peripheral tissues – highlighting a requirement for microglial-intrinsic parasite control. Importantly, we

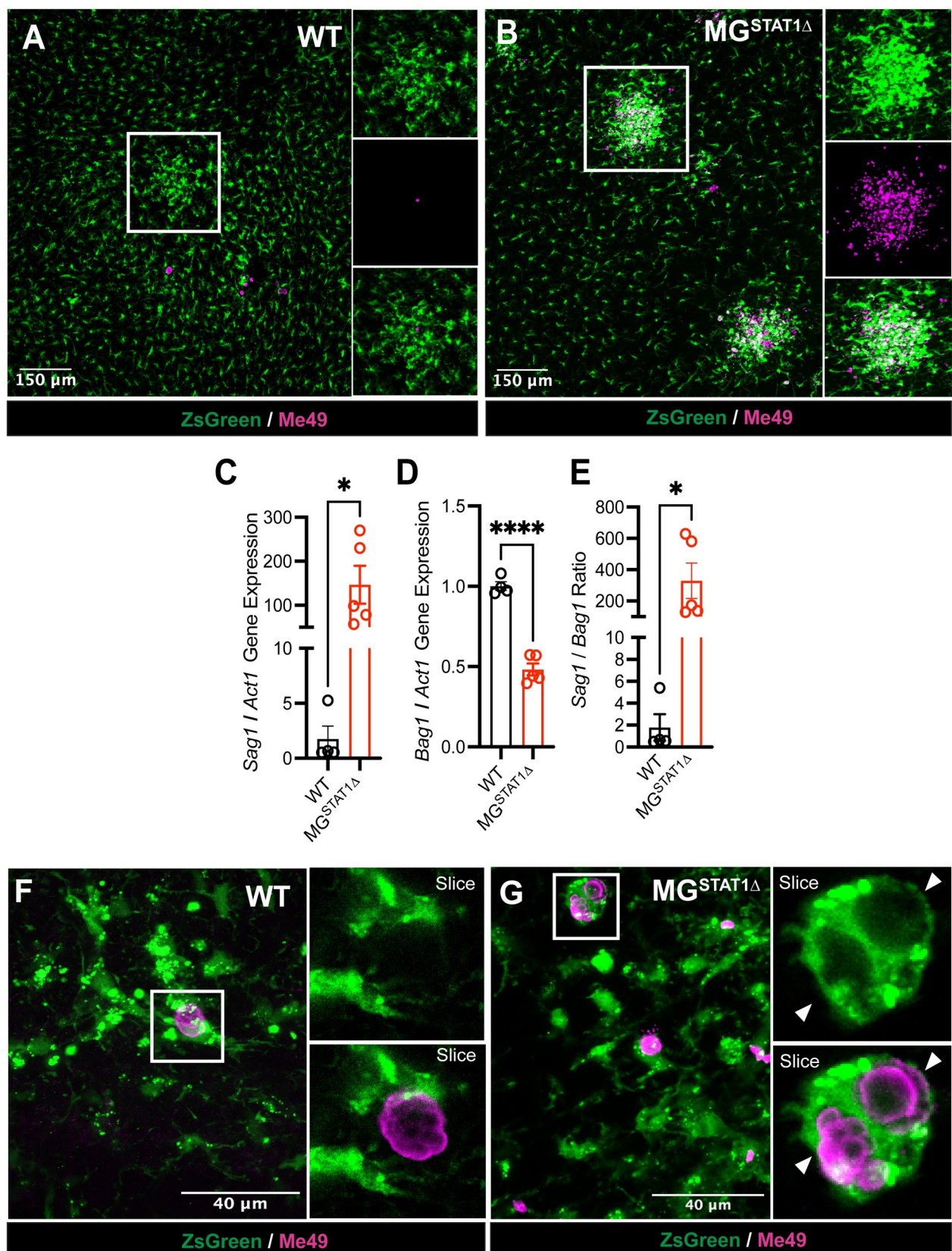

**Fig 6. Microglial STAT1-deficiency results in a skewing of *T. gondii* toward its replicative form.** WT and MG$^{STAT1\Delta}$ mice were intraperitoneally infected with 10 cysts of the Me49 strain of *T. gondii* and brains were analyzed by confocal microscopy and RT-qPCR at 15 DPI. (**A-B**) Representative 10x immunofluorescent confocal Z-stack images of *T. gondii* inflammatory foci in infected brains. White square indicates example foci, with insets providing zoomed detail. (**C-E**) RT-qPCR analysis of *T. gondii* relative gene expression of *Sag1* (tachyzoite-specific gene), *Bag1* (bradyzoite-specific gene), and *Act1* (non-stage-specific gene) analyzed using *t* tests. (**F-G**) Representative 40x Immunofluorescent images of microglial-parasite interactions in infected brains. White square indicates *T. gondii* vacuole(s) in maximum projection view, with insets providing zoomed detail in single-plane view (Slice). (**A-B**) and (**F-G**) ZsGreen is indicated green, and anti-Me49 staining is indicated in magenta. Scale bar = 150 μm (**A-B**) and 40 μm (**F-G**). *n* = 3-4 mice per group (**C-E**). ns = not significant, * = $p < 0.05$, *** = $p < 0.001$.

show that contrary to interpretations from previous literature [44], microglia do not express iNOS during *in vivo T. gondii* challenge. Instead, transcriptomic analysis suggests that a loss of STAT1-regulated cytosolic killing genes, including those from the IRG and GBP superfamilies, may normally play a role in preventing microglia from serving as a cellular niche in which *T. gondii* can freely replicate. Consistent with this model, we observe that STAT1-deficiency in microglia results in an increased brain parasite burden and a skewing of parasite towards its fast-replicating and lytic tachyzoite form within the brain.

As with any study, there are caveats and limitations that must be appropriately considered in interpreting the presented data. Importantly, the CX3CR1$^{CreERT2}$ system allows for substantial turnover of short-lived circulating immune cells [24,39] but is expected to additionally target other tissue-resident macrophage populations [39,63]. Experiments were performed to verify effective parasite clearance and immunity in peripheral tissues, but there exists a relatively small but long-lived CX3CR1+ population of border-associated CNS macrophages (BAMs), located along CNS interfaces [63]. We expect that tamoxifen treatment may yield STAT1-deficient BAMs, such as perivascular and choroid plexus macrophages, which may be spatially positioned to encounter *T. gondii* before it breaches into the brain parenchyma. While we did not observe increased parasite burden at 8 DPI, when parasite is seeding the brain between MG$^{STAT1\Delta}$ and WT mice, future experiments are necessary to determine whether BAMs play an active role in parasite restriction, and whether these roles are similar to those of microglia.

While STAT1 is activated downstream of multiple interferons and IFN-γR activation was not directly investigated in the current study, multiple lines of evidence support the hypothesis that a defect in the ability for microglia to respond to IFN-γ is the primary driver of the pathology observed in MG$^{STAT1\Delta}$ mice during *T. gondii* challenge. IFN-γ is essential during both acute and chronic *T. gondii* infection, with STAT1-/- mice closely phenocopying IFN-γ-/- mice in uniformly succumbing during the acute phase of infection [4], and pharmacological depletion of IFN-γ specifically during chronic infection similarly resulting in uniform lethality [5]. In contrast to IFN-γ, IFN-α/β appear to carry a more limited protective role during *T. gondii* infection [4, 64]. In an oral model of infection, global genetic deletion of their shared receptor, IFNAR, results in increased brain parasite burden but more modest levels of animal mortality (~50% mortality in IFNAR-/- vs. ~20% mortality in wild-type controls by 50 DPI) [64]. Together, this literature supports a predominant role for IFN-γ-STAT1 signaling in the current study. However, future experiments will be necessary to determine to what extent IFN-α/β, or other potential activators of STAT1 such as Il-27 and Il-6 [65–68], could play compounding roles or are independently capable of shaping the microglial response to *T. gondii* infection. Lastly, our RNA-sequencing data did not suggest appreciable levels of IFN-λR on microglia, suggesting limited relevance of type III interferons to this cell type [data publicly available through GEO accession code GSE203655].

Our finding that STAT1-deficiency in microglia leads to a 300-fold increase, relative to WT controls, in the replicative tachyzoite form of *T. gondii*, is well-framed within the context of a previous study that examined the role of STAT1 signaling in astrocytes [69]. We observe

multiple similarities between our data and this study: (i) increased animal mortality, (ii) increased brain parasite burden, (iii) liquefactive necrosis throughout the brain, and (iv) a loss of IRG and GBP expression with concomitant susceptibility of the targeted STAT1-deficient cell type to *T. gondii* parasitization. Our study dovetails with this previous publication in illustrating the essential role of STAT1 signaling in both hematopoietic and non-hematopoietic cell types, and suggests that many different cell types may be able to serve as replicative niches for this opportunistic pathogen in the absence of STAT1. However, there are important differences observed between microglial and astrocytic STAT1 deletion during *T. gondii* infection that may be informative in better understanding cell type-parasite interactions.

First, the animal mortality observed in MG$^{STAT1\Delta}$ mice in our study occurs more rapidly (~17 DPI) than as reported with astrocytic STAT1-deletion (~25 DPI), and with greater penetrance, indicating more severe disease pathology [69]. Second, MG$^{STAT1\Delta}$ mice display increased, rather than equivalent, T cell and infiltrating myeloid functional responses [69]. Third, astrocytic STAT1-deletion appears to promote the bradyzoite form of *T. gondii*, whereas MG$^{STAT1\Delta}$ mice show strong bias toward the tachyzoite form of *T. gondii* [69]. Different physiological properties of these two glial populations may offer insights into these key differences in phenotype. Microglia, relative to astrocytes, are highly motile cells with a complex sensome that drives chemotactic responses to tissue damage or disruption [70–73], as is evident in our confocal images of microglia abandoning their evenly-tiled territories and clustering around *T. gondii* tachyzoites at multiple time-points, regardless of STAT1-sufficiency. Rapid migration of STAT1-sufficient microglia to sites of lytic tachyzoite egress may thus position these cells to serve as an immunological "cellular buffer" – becoming actively infected and subsequently clearing intracellular infection via STAT1-dependent cytosolic killing molecules. In the absence of STAT1 signaling, this chemotactic response may contrastingly permit the parasite to be more rapidly passaged through a hospitable cell type that is unable to mount a cell-intrinsic immune response, thus potentially overwhelming brain tissue. Future studies will be needed to understand if the observed skewing of parasite form toward its tachyzoite state reflects an inability for tachyzoites to convert to bradyzoites due to intrinsic properties of microglia, or alternatively reflect an interaction of STAT1 signaling abrogation with unknown biological factors of early-chronic infection (15 DPI).

While the precise determinants of tachyzoite relative to bradyzoite enrichment in MG$^{STAT1\Delta}$ mice remain unclear, the observation that brain-infiltrating T cell and myeloid responses are increased in MG$^{STAT1\Delta}$ relative to control mice may best be explained as a consequence of increased brain parasite burden, rather than a direct result of microglial STAT1 deletion. We expect that increased parasite burden stemming from impaired microglia-mediated tachyzoite killing would lead to increased T cell stimulation due to an increase in antigen availability in the brain that can be presented by STAT1-competent infiltrating myeloid cells. Consistent with this view, we observed an increase in *T. gondii* antigen-specific CD4+ T cells in the brain, measured by flow cytometry using an MHC II tetramer, as well as increased CD4 + and CD8+ T cell IFN-γ production. Moreover, increased parasite replication and egress, a lytic process [reviewed in ref. 74], may also increase the release of inflammatory signals such as alarmins that drive further recruitment of immune cells from circulation [11, 75]. The inability of a robustly activated brain-infiltrating immune compartment to compensate for microglial STAT1 deficiency thus underscores the necessity for STAT1 signaling in brain-resident populations during *T. gondii* challenge.

Our data also provide multiple insights to a broader understanding of microglial identity and their physiology across disease states. While most macrophages typically display remarkable plasticity attuned to their tissue microenvironments [8,76,77], the observation that microglia do not express iNOS, despite sharing an interferon-primed environment with iNOS

+ infiltrating myeloid cells, indicates that these two cell populations provide differential immune effector functions during *T. gondii* infection. This differential expression of immune effector mechanisms mirrors previous work from our lab showing that microglia, unlike infiltrating macrophages: (i) express low levels of IL-1β during chronic infection, and (ii) display a comparatively dampened NF-κB signature during *T. gondii* challenge [11]. In the present study, we also find that microglia acquire a neurodegenerative-like DAM signature during *T. gondii* infection, and that this signature is partially regulated by STAT1 signaling during parasitic infection. While caution must be exercised in attributing the DAM phenotype to parasite restriction, this gene expression signature highlights an inability of STAT1-deficient microglia to acquire the cell type-specific transcriptional hallmarks of immune activation conserved across other neuroinflammatory models. Future studies targeting genes such as TREM2 or APOE, which have been shown to regulate the microglial transition to DAMs, may be needed to discern whether DAMs are, in themselves, neuroprotective during CNS infection. Similarly, future experiments using MG$^{STAT1\Delta}$ mice in models of neurodegeneration may additionally yield novel insights into how the microglial response to interferon shapes or mitigates disease progression in a wide range of disease models.

## Materials and methods

### Ethics statement

All procedures involving animal care and use were approved by and conducted in accordance with the University of Virginia's Institutional Animal Care and Use Committee (IACUC) under protocol number 3968.

### Mice

CX3CR1$^{CreERT2}$ (#020940) and ROSA26$^{Ai6/Ai6}$ (#007906) mouse lines were originally purchased from Jackson Laboratories and cross-bred to generate CX3CR1$^{CreERT2}$ x ZsGreen$^{fl/stop/fl}$ mice used as controls. These control mice were subsequently cross-bred with STAT1$^{fl/fl}$ mice (provided by Lothar Hennighausen, NIH) to generate STAT1$^{fl/fl}$ x CX3CR1$^{CreERT2}$ x ROSA26$^{Ai6/Ai6}$ (MG$^{STAT1\Delta}$) mice. Age- and sex-matched mice were intraperitoneally administered tamoxifen (4 mg per 20 g body weight, Sigma-Aldrich) between 4-7 weeks of age for 5 consecutive days to induce STAT1 deletion. Four weeks following tamoxifen treatment, mice were challenged with the type II *T. gondii* strain Me49 or proceeded with naïve experiments. Parasite was passaged through CBA/J and Swiss Webster mice (Jackson Laboratories), and mice used in infection experiments were intraperitoneally challenged with 10 Me49 cysts from CBA/J brain homogenate. *Stat1* excision was confirmed by qPCR, flow cytometry, or immunohistochemistry. All mice were housed in University of Virginia specific pathogen-free facilities with a 12h light/dark cycle, with ambient temperature between 68 and 72 F, and 30-60% humidity. Mice used in experiments were euthanized by $CO_2$ asphyxiation if they showed weight loss greater than 20% of their baseline, pre-recorded weight.

### Immunohistochemistry

PBS-perfused brain hemispheres were fixed in 4% paraformaldehyde (Electron Microscopy Sciences) for 24 hours at 4˚C. Brains were then cryopreserved in a 30% sucrose solution for 48 hours at 4˚C, flash-frozen in OCT (Sakura), on dry ice, and cryosectioned at a thickness of 50 μm. Free-floating sections were blocked for 1 hour at room temperature in a 1x PBS solution containing 0.1% Triton, 0.05% Tween 20, 5% BSA, and 0.1% BSA. Sections were incubated in primary antibodies overnight at 4˚C. Sections were washed with a 1x PBS, 0.05% Tween solution prior to

incubation in primary antibodies for one hour at RT. Following secondary washes, sections were incubated in DAPI (Thermo Fisher Scientific) for 5 minutes and mounted on glass coverslips (Thermo Fisher Scientific) in Aquamount (Lerner Laboratories) mounting media. Dilutions were performed at a 1:200 concentration for primary antibodies and a 1:1,000 concentration for secondary antibodies and DAPI staining. All immunohistochemical micrographs were captured using a Leica TCS SP8 Confocal microscope. Images were analyzed using Imaris or ImageJ software. Primary antibodies used in experiments included goat anti-Iba1 (Abcam), rabbit anti-Me49 (gift from Fausto Araujo), and rat anti-MHC II (Cat. #: 14-5321-82; Thermo Fisher Scientific). Secondary antibodies used in experiments included: Alexa Fluor 594 Donkey Anti Rabbit IgG (Cat. #: 711-585-152; Jackson ImmunoResearch), Alexa Fluor 647 Donkey Anti-Rabbit IgG (Cat. #: 711-605-152; Jackson ImmunoResearch), Cy3 AffiniPure Donkey Anti Rat IgG (Cat. #: 712-165-153; Jackson ImmunoResearch).

## Sholl analysis

Sholl analysis was used to analyze microglial morphological complexity as a readout for their activation state, in accordance with the protocol published in *Norris et al., 2014* [40]. In brief, confocal photomicrographs of naive WT and MG$^{STAT1\Delta}$ brains with ZsGreen+ microglia were captured and analyzed in Fiji with the Sholl Analysis plugin software. Images were processed in binary pixels, microglia were manually selected, and shells were inserted in 5 μm concentric circles, starting from 10 μm outside the center of the soma, and ending at the limit of the longest arborization for each analyzed microglia. The number of dendritic intersection points was plotted for each distance away from the soma, and data was analyzed via 2-way ANOVA with Sidak's multiple comparison test.

## H&E tissue staining

Brains were fixed in formalin, prior to being embedded in paraffin, sectioned, and stained with hematoxylin and eosin at the UVA Research Histology Core. Following mounting, sections were imaged on a DM 2000 LED brightfield microscope (Leica Biosystems).

## Tissue processing

Mice were given an overdose of ketamine/xylazine and transcardially perfused with 30 mL of ice-cold 1X PBS. Brains and spleen were collected and placed in cold complete RPMI media (cRPMI) (10% FBS, 1% sodium pyruvate, 1% non-essential amino acids, 1% penicillin/streptomycin, and 0.1% 2-ME). Brains were minced with a razor blade, passed through an 18G needle, and enzymatically digested with 0.227 mg/mL collagenase/dispase and 50U/mL DNase (Roche) for 45 minutes at 37C. If brain samples were used to quantify parasite burden or for gene expression analysis, aliquots were removed and frozen for downstream analysis prior to addition of digestion enzymes. If brain cells were being used for downstream RNA sequencing for WT vs. MG$^{STAT1\Delta}$ analysis, Actinomycin D (Sigma-Aldrich) was added to the digestion buffer at a concentration of 45 μM during incubation and 3 μM during washes to inhibit upregulation of immediate early activation genes associated with subsequent FACs sorting. All brain samples were resuspended in 20 mL of 40% percoll and spun for 25 minutes at 650 xg to remove myelin. Following myelin removal, samples were washed with and subsequently resuspended in cold cRPMI. Blood collected for flow cytometric analysis was isolated from the heart prior to transcardial perfusion and transferred into 1x PBS + EDTA (Thermo Fisher Scientific). Blood was then processed with RBC lysis and resuspended in cold cRPMI prior to staining and fixation for flow cytometry. For peritoneal lavage experiments, 5 mL of cold 1X

PBS was injected through the membrane encasing the peritoneal cavity via a 26G needle and withdrawn with a 22G needle. Lavage fluid was washed and suspended in cRPMI.

## Flow cytometry

Following generation of a single-cell suspension, cells were plated in a 96-well plate and incubated for 10 minutes in 50 μL Fc block (1 μg/mL 2.4G2 Ab (BioXCell), 0.1% rat γ-globulin (Jackson ImmunoResearch) at room temperature. Cells were incubated in primary antibodies or tetramer at a concentration of 1:200, and AF-780 viability dye (eBioscience) at a concentration of 1:800 for 30 minutes at 4°C. Antibody clones used for experiments included: MHC II (M5/114.15.2), CD11b (M1/70), Ly6C (HK1.4), iNOS (CXNFT), CD45 (30-F11), CD3e (145-2C11), CD4 (GK1.5), CD8a (53-6.7), and IFN-γ (XMG1.2) (Thermo Fisher Scientific). In experiments analyzing antigen-specific CD4+ T cell responses, cells were pre-incubated with PE-conjugated MHC II I-A$^b$ AS15 tetramer (NIH Tetramer Core Facility) for 15 minutes at room temperature before addition of antibodies to the staining buffer. After staining for surface markers, cells were washed and fixed overnight in 2% PFA at 4°C, before being washed and intracellularly stained, if quantifying cytosolic protein. For intracellular cytokine staining (IFN-γ), initial single cell suspensions were incubated with Brefeldin A (Selleckchem) for 5 hours at 37°C prior to blocking and staining. For intracellular cytokine analysis, cells did not receive any additional stimulation *ex-vivo*. For intracellular staining, cells were permeabilized with Permeabilization Buffer (eBioscience) and stained for 30 minutes at room temperature. Cells were washed with FACS buffer and transferred into 5 mL FACS tubes, then were analyzed on a Gallios flow cytometer (Beckman-Coulter). Flow cytometry data was analyzed using FlowJo.

## Cell sorting / Enrichment

For RNA sequencing and analysis of microglial *Stat1* relative expression to validate excision, brains were processed into a single cell suspension, as described above. Cells were then magnetically labeled with CD11b-conjugated beads diluted in MACS buffer for 15 minutes, per manufacturer's instructions. Following a wash with 2 mL of MACS buffer, samples were spun at 1500 RPM for 5 minutes and resuspended in 600 uL of MACS buffer. Myeloid cells were then positively selected for using anti-CD11b-conjugated magnetic beads enrichment (Miltenyi). Cells were resuspended and lysed in Trizol for RT-qPCR analysis, or incubated for 10 minutes in 50 μL Fc block (1 μg/mL 2.4G2 Ab (BioXCell), 0.1% rat γ globulin (Jackson ImmunoResearch) at room temperature for RNA-sequencing. Cells were stained with the following antibodies (Thermo Fisher Scientific) for 30 minutes at 4°C: CD11b-Percp Cy5.5 (Cat. #45-0112-82), MHCII-eFluor 450 (Cat. # 48-5321-82), CD45-APC (Cat. #17-0451-81), Ly6C-PE Cy7 (Cat. #12-5932-80), CD3e-PE Cy7 (Cat. #25-0031-81), NK1.1-PE Cy7 (Cat. #25-5941-81), CD19-PE Cy7 (Cat. #25-0193-81). Live cells were analyzed and sorted using a BD Aria flow cytometer at the University of Virginia Flow Cytometry Core facility. Cells were sorted based on ZsGreen and dump gating (CD3e$^-$ NK1.1$^-$ CD19$^-$ Ly6C$^-$) directly into Trizol (Invitrogen) for RNA extraction and RNA sequencing. For MG$^{STAT1Δ}$ mice, MHC II$^{neg}$ microglia were gated in order to positively select for STAT1-deficient cells.

## Quantitative RT-PCR

For tissue-level analysis, one-fourth of a mouse brain was placed in 1 mL Trizol (Ambion), mechanically homogenized using 1 mm zirconia/silica beads (Biospec) for 30 seconds using a Mini-BeadBeater 16 (BioSpec). For gene expression analysis of magnetically-enriched cells, cells were homogenized in Trizol by pipetting. RNA was extracted from Trizol according to manufacturer's instructions (Invitrogen). High Capacity Reverse Transcription Kit (Applied

Biosystems) was used to generate cDNA. Quantitative PCR was performed using 2X Taq-based Master Mix (Bioline) and TaqMan gene expression assays (Applied Biosystems), or custom primers (Integrated DNA Technologies), run on a CFX384 Real-Time System thermocycler (Bio-Rad Laboratories). Murine *Hprt* and *T. gondii Act1* were used for normalization for analyzing host and parasite gene expression, respectively, and relative expression is reported as $2^{(-\Delta\Delta CT)}$. The following Thermo Fisher mouse gene probes were used: *Stat1* (Mm00439518_m1*)*, *Hprt* (Mm00446968_m1), *Ifng* (Mm01168134_m1), *Nos2* (Mm00440502_m1), *Il6* (Mm00446190_m1), *Icam1* (Mm00516023_m1), *Vcam1* (Mm01320970_m1), *Tnfa* (Mm00443258_m1), *Ccl2* (Mm00441242_m1), *Ccl5* (Mm01302427_m1), *Cxcl9* (Mm00434946_m1), *Cxcl10* (Mm00445235_m1). Custom primers for used for analyzing *T. gondii* genomic DNA and gene expression were used and are provided in **S1 Table**.

## RNA sequencing analysis

RNA reads from FASTQ files were trimmed and filtered using Trimmomatic (v0.39) paired-end set to phred 33 quality scoring. Adapters were trimmed, and reads with a minimum quality score of 15, leading and trailing quality scores of 3, and minimum fragment length of 36 were used for analysis. FastQC (v0.11.9) was used to verify quality of sample reads. Trimmed and filtered reads were aligned to thee GENCODE M13 reference genome using Salmon (v0.8.2) and output as sam files. Transcript abundance files were imported into R (v4.1.1) and converted to gene abundances using Tximport (v1.24.0). The R Bioconductor package, DESeq2 (v1.36.0), was used to perform differential expression analysis. DESeq2-normalized data was visualized using the following R packages: EnhancedVolcano (v1.14.0), pHeatmap (v1.0.12), and ggplot2 (v3.3.6). Gene names were converted from mouse ENSEMBL gene identifiers to gene symbols using the Bioconductor BiomaRT (v2.52.0) database. Labeled genes were manually selected from significantly differentially expressed genes from the DESeq2 results data frame. All genes with a Benjamini-Hochberg (BH) adjusted p-value below 0.05 were considered significantly upregulated if they had a log2FC > 0, and downregulated if they had a log2FC < 0. In order to determine Gene Ontology (GO) enrichment analysis for biological processes was performed by running a list of significantly differentially expressed genes and their p-values through TopGO (v2.48.0). Enrichment score is reported as the $-\log_{10}$ of enrichment p value, based on Kolmogorov-Smirnov (KS) analysis. Significantly enriched GO Terms were selected and plotted based on biological interest. For analysis of the DAM signature, the full list of common genes upregulated (disease-associated microglia genes) and downregulated by microglia (homeostatic genes) across disease conditions in *Krasemann et al.*, *2017* was analyzed in our dataset, and significantly differentially expressed genes from this list were plotted. For targeted analysis of anti-parasitic genes, the full list of unique annotated genes in the GO:0042832 Term ("Defense Response to Protozoan") was analyzed within our dataset, and all significantly differentially expressed genes were plotted.

## Statistics

Data from multiple experiments within a given time-point were aggregated to reflect biological variability from different infections. When data from multiple infection cohorts were compiled, a randomized block ANOVA test (two-way) was performed in R. This statistical test was selected to evaluate the effect of treatment group, while controlling for infection date as a variable that was statistically modeled as a random effect contained in our datasets. Data from Kaplan-Meier curves, flow cytometric analyses, and qPCR results were graphed using Graph-Pad Prism, and data related to transcriptomic analysis were graphed using R. Error bars

indicate standard error of the mean (SEM). Statistical tests used for each reported experiment are detailed within figure legends. Normal distributions for each population are assumed for statistical testing.

## Dryad DOI

https://doi.org/10.5061/dryad.fttdz08w2 [78]

## Supporting information

**S1 Fig. Example gating strategy for brain immune cells.** Myeloid and T cells isolated from mice with ZsGreen+ microglia were analyzed via flow cytometry. (**A-C**) For all panels, cells were pre-gated on singlets (**A**), then live cells using a viability dye (**B**). (**C**) Cells were gated to identify CD45 hi (brain-infiltrating) and CD45 int (brain-resident) immune cells. (**D**) Microglia were gated based on CD45 intermediate expression, ZsGreen and CD11b expression, and (**E-F**) MHC II positivity was assessed using FMO. (**G**) Infiltrating myeloid cells were gated based on CD45 hi expression, and the expression of both CD11b and Ly6C. (**H-I**) iNOS expression was determined via FMO gating. (**J-K**) T cells were gated based on the expression of CD3 and CD4 or CD8. (**L-M**) IFN-γ expression on CD3+CD4+ and CD3+CD8+ cells was determined based on FMO positivity.
(TIF)

**S2 Fig. RNA sequencing analysis of naïve vs. *T. gondii*-infected WT microglia.** Microglia from wild-type naïve or wild-type mice infected with *T. gondii* for 4 weeks were FACS-sorted and RNA-sequenced at 4 weeks post-infection. (**A-B**) Heat maps displaying hierarchically-clustered gene expression from regularized log transformed gene abundance counts. Heatmap data display the full set of significantly differentially expressed microglial homeostatic genes (**A**), or disease-associated microglia genes (**B**), shared across neurodegenerative models investigated in *Krasemann et al.*, *2017* and reflected in the naïve vs. T. *gondii*-infected DESeq2 dataset. Statistical significance was defined in the differential gene expression analysis as a BH adjusted *p* value < 0.05. *n* = 4-5 mice per group.
(TIF)

**S3 Fig. Validation of cre activity and STAT1 excision in MG$^{STAT1\Delta}$ mice.** Naïve microglia and microglia isolated from brains at 12 DPI were analyzed by flow cytometry or RT-qPCR for relative gene expression. (**A-B**) Representative FACS plots indicating gating strategy for validating microglial ZsGreen expression in naïve mice. (**C**) Flow cytometric quantification of ZsGreen expression in total CD45$^{int}$ CD11b+ cells in naïve WT or MG$^{STAT1\Delta}$ mice. (**D**) RT-qPCR quantification of *Stat1* relative expression in microglia that were magnetically enriched from naïve vehicle or tamoxifen (TAM)-treated MG$^{STAT1\Delta}$ mice. (**E**) Flow cytometric quantification of ZsGreen expression in various immune populations isolated from blood at 12 DPI, in WT or MG$^{STAT1\Delta}$ mice. Statistical significance was determined via unpaired *t* test, with *n* = 3-5 mice per group (**C-E**). ns = not significant; **** = $p < 10^{-4}$.
(TIF)

**S4 Fig. *T. gondii* burden in peripheral tissues in WT and MG$^{STAT1\Delta}$ mice.** WT and MG$^{STAT1\Delta}$ mice were intraperitoneally infected with 10 cysts of the Me49 strain of *T. gondii*, and peripheral tissues were harvested and analyzed by qPCR for parasite genomic DNA, relative to total tissue DNA. Parasite burden was quantified at 8 DPI in lung (**A**), liver (**B**), and heart (**C**) tissue. Parasite burden was quantified at 15 DPI in lung (**D**), liver (**E**), and heart (**F**) tissue. (**A-C**) Statistical significance was determined via randomized block ANOVA using compiled data from 2-3 experiments with *n* = 10 mice per group (**A**), *n* = 7-8 mice per group

(**C**), n = 9-11 mice per group (**D-F**), or via unpaired t test with $n$ = 3-4 mice per group (**B**). Dotted line on y axis denotes assay limit of quantification, based on lower limit of standard curve. ns = not significant.
(TIF)

**S5 Fig. WT and MG$^{STAT1\Delta}$ mice display equivalent immune activation in peripheral tissues during *T. gondii* challenge.** WT and MG$^{STAT1\Delta}$ mice were intraperitoneally infected with 10 cysts of the Me49 strain of *T. gondii*, and immune cells from blood and peritoneal fluid were analyzed by flow cytometry. (**A-E**) Flow cytometric quantification of total live immune cells (**A**), CD3+CD4+ T cell count (**B**), CD3+CD8+ T cell count (**C**), CD11b+Ly6Chi monocytes (**D**), and CD11b+Ly6Clo monocytes (**E**), calculated from blood. (**F-J**) Quantification of total live cells (**F**), number of CD3+CD4+ T cells, (**G**) number of CD3+CD8+ T cells (**H**), and CD4 + or CD8+ T cell expression of IFN-γ (**I-J**) isolated from the peritoneal cavity at 8 DPI. Statistical significance was determined by two-way randomized block ANOVA (**A-J**). ns = not significant, $n$ = 11 per group from two pooled experiments (**A-E**), or $n$ = 7-9 mice per group from two pooled experiments (**F-J**). Biological replicates are individual mice, with group means from individual experiments plotted as open circles with black lines connecting experimental and control groups. Source data (**A-H**) are provided in a source data file.
(TIF)

**S6 Fig. Morphometric analysis of WT and STAT1-deficient microglia.** To analyze microglial activation, Sholl analysis was performed on microglia from the somato-motor cortex of naïve WT and MG$^{STAT1\Delta}$ mice. (**A**) ZsGreen+ microglia were imaged using a confocal microscope, and images were processed into a maximum projection using Fiji. (**B**) Images were made binary, microglia were manually isolated to determine cell process continuity, and the Sholl analysis Fiji plugin was executed to record intersections at varying soma distances. (**C**) Quantification of Sholl data via two-way ANOVA with Sidak's multiple comparisons test, $n$ = 64-67 microglia from 3 mice per group. (**A**) Scale bar = 40 μm.
(TIF)

**S7 Fig. Microglial antigen presentation machinery is regulated by STAT1 during *T. gondii* infection.** Microglia were isolated from naïve or 12 DPI infected WT or MG$^{STAT1\Delta}$ mouse brains and analyzed by flow cytometry and confocal microscopy. Flow cytometric analysis of microglial major histocompatibility complex and co-stimulatory molecules in (**A**) naïve brains, and (**B**) 12 DPI brains. (**C-D**) Immunohistochemical analysis of MHC II positivity by confocal microscopy; scale bar = 40 um, blue indicates MHC II, and green indicates ZsGreen fluorescence. $n$ = 2-3 per group, unpaired *t* test (**A-B**). ns = not significant, ** = $p$ <0.01, *** = $p < 0.001$.
(TIF)

**S8 Fig. Antigen-specific CD4+ T cell responses are increased in MG$^{STAT1\Delta}$ mice.** WT and MG$^{STAT1\Delta}$ mice were intraperitoneally infected with 10 cysts of the Me49 strain of *T. gondii*, and brain-infiltrating CD4+ T cells were analyzed by flow cytometry for MHC II I-A$^b$ AS15 tetramer positivity at 15 DPI. (**A-B**) Representative FACS plots indicating tetramer gating in WT (**A**) and MG$^{STAT1\Delta}$ mice (**B**). (**C**) Quantification of CD3+CD4+ tetramer+ cells isolated from brains at 15 DPI via unpaired *t* test, $n$ = 3 and 5 mice per group. * = $p$ <0.05. Error bars indicate standard error of the mean.
(TIF)

**S9 Fig. Whole brain RNA analysis of anti-parasitic defense genes.** Whole brain homogenate from WT and MG$^{STAT1\Delta}$ mice was analyzed by RT-qPCR for a panel of various anti-parasitic

genes. (**A**) 12 DPI and (**B**) 15 DPI immune effector profile. Statistical significance was determined by unpaired $t$ test for one experiment (**A**), and two-way randomized block ANOVA from two pooled experiments (**B**). $n = 8$ per group (**A**), and $n = 9$-10 mice per group (**B**). $^* = p < 0.05$, $^{**} = p < 0.01$, $^{***} = p < 0.001$, $^{****} = p < 10^{-4}$. Error bars indicate standard error of the mean.
(TIF)

**S1 Table. List of qPCR primers and probes used for analyzing *T. gondii* genomic DNA and RNA gene expression.**
(PDF)

## Acknowledgments

The authors would like to thank members of the Harris Lab, Center for Brain Immunology and Glia (BIG), Carter Immunology Center (CIC), and Neuroscience Graduate Program at the University of Virginia for their input in the development of this work. We also would like to thank Will Rosenow, Marieke Jones, PhD, and Christopher Overall, PhD for their assistance with the computational analysis of RNA-sequencing data, as well as the UVA Flow Cytometry Core, Research Histology Core, Health Sciences Library, and Research Computing for methodological assistance. We thank Lothar Hennighausen (NIH) for gifting us STAT1<sup>fl/fl</sup> mice that were used to generate experimental mice, and Fausto Araujo (Palo Alto Medical Foundation) for gifting us the anti-Me49 antibody used in experiments. Parts of Figs 1 and 2 were created with Biorender.com.

## Author Contributions

**Conceptualization:** Maureen N. Cowan, Michael A. Kovacs, Samantha J. Batista, Tajie H. Harris.

**Data curation:** Maureen N. Cowan, Jeremy A. Thompson.

**Formal analysis:** Maureen N. Cowan.

**Funding acquisition:** Tajie H. Harris.

**Investigation:** Maureen N. Cowan, Ish Sethi, Isaac W. Babcock, Katherine Still, Samantha J. Batista, Jeremy A. Thompson.

**Methodology:** Maureen N. Cowan, Katherine Still, Carleigh A. O'Brien, Tajie H. Harris.

**Project administration:** Tajie H. Harris.

**Supervision:** Katherine Still, Carleigh A. O'Brien, Tajie H. Harris.

**Validation:** Maureen N. Cowan.

**Visualization:** Maureen N. Cowan.

**Writing – original draft:** Maureen N. Cowan.

**Writing – review & editing:** Maureen N. Cowan, Michael A. Kovacs, Ish Sethi, Isaac W. Babcock, Lydia A. Sibley, Sydney A. Labuzan, Tajie H. Harris.

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
