## [Decision Letter · Decision Letter 0]

20 Jul 2022

Dear Dr. Harris,

Thank you very much for submitting your manuscript "Microglial STAT1-sufficiency is required for resistance to toxoplasmic encephalitis" for consideration at PLOS Pathogens. As with all papers reviewed by the journal, your manuscript was reviewed by members of the editorial board and by several independent reviewers. The reviewers appreciated the attention to an important topic. Based on the reviews, we are likely to accept this manuscript for publication, providing that you modify the manuscript according to the review recommendations.

Both reviewers were in agreement about the importance of this study with new insight into STAT1 signaling in microglia, well-supported conclusions, and robust experimental design. Minor comments are provided that should be addressed by the authors.

Sincerely,

Meera Goh Nair

Associate Editor

PLOS Pathogens

Kami Kim

Section Editor

PLOS Pathogens

Kasturi Haldar

Editor-in-Chief

PLOS Pathogens

orcid.org/0000-0001-5065-158X

Michael Malim

Editor-in-Chief

PLOS Pathogens

orcid.org/0000-0002-7699-2064

Both reviewers were in agreement about the importance of this study with new insight into STAT1 signaling in microglia, well-supported conclusions, and robust experimental design. Minor comments are provided that should be addressed by the authors.

Reviewer Comments (if any, and for reference):

Reviewer's Responses to Questions

**Part I - Summary**

Reviewer #1: Here the authors examine the role of microglia in protecting against Toxoplasma gondii infection in the CNS. Specifically, authors use a conditional genetic ablation strategy to eliminate STAT1 signaling specifically in microglia. Results show that STAT1 signaling in microglia is critical for the transition of microglia from homeostatic to a “disease-associated” phenotype as well as upregulation of anti-parasitic genes. Interestingly, this does not appear to negatively impact the function of infiltrating immune cells. Genetic deletion of STAT1 in microglia results in overwhelming T. gondii replication in the CNS and ultimately, uniform lethality of infected mice. One important aspect of these data is that they show that what the field refers to as a “disease-associated microglia” phenotype is, in fact, important for CNS immunity against pathogens. Overall, this article is well-written and the conclusions are well-supported by the data. I have only minor critiques for the authors to clarify before it is acceptable for publication.

Reviewer #2: In this study, Cowan et al. provide an important and robust study of the contributions of microglial STAT1 signaling in controlling neuroinvasive T. gondii infection. The authors use genetic strategies to specifically delete STAT1 from microglia without altering STAT1 in myeloid cells of bone-marrow origin, which they extensively validate with multiple approaches. Microglial STAT1 appears to be absolutely essential for control of T gondii infection in the brain, and the authors provide several important insights into the protective mechanisms elicited by STAT1 signaling in this cell type. Remarkably, the authors show that, despite an overall “inflammatory” gene signature, microglia simply do not express iNOS, which is restricted exclusively to infiltrating myeloid cells. Further, STAT1 signaling in microglia is associated with the conferral of a “DAM” like transcriptional program. These findings challenge the prevailing paradigm that focuses on the neuropathogenic roles of microglial activation in sterile neuroinflammatory disorders, and begins to illuminate the likely evolutionary purpose of “DAMs,” if such a cell type can be said to exist. While much remains to be done in future work, the description of antimicrobial functions associated with the DAM signature is an important finding. Overall, this manuscript was a pleasure to review. It is well written, boasts extensive and convincing controls throughout, and is circumspect in the interpretation of its findings.

**Part II – Major Issues: Key Experiments Required for Acceptance**

Reviewer #1: No major issues identified.

Reviewer #2: I do not identify any experiments that are absolutely required for publication. The authors have done an exemplary job of performing extensive controls, and have been reasonably tempered conclusions in the occasional absence of extensive data.

**Part III – Minor Issues: Editorial and Data Presentation Modifications**

Reviewer #1: 1. Materials and Methods, Line 475: Is it correct that brains were embedded in formalin?

2. Materials and Methods, Line 489: Authors refer to the concentration of Actinomycin D in digestion buffer as 45 um and 3 um. I believe this should probably be uM.

3. Materials and Methods, Line 550: Authors refer to Supplemental Table 1, which I am not seeing included in the Supplemental File.

4. Figure Legend 1: Please include number of mice per group that were analyzed. In other figure legends, the authors list the “total n”. The format in which the data is presented it is impossible to tell how balanced these groups are, and a number per group (or even a range) would be more informative (ie, n=6-7 per group rather than n=13).

5. Figure 2: What is the time point post-infection for panel 2B?

6. Please note that the legend for Fig 6 is duplicated in the Supplemental Information.

7. Supplemental Figure 3: is it correct that the flow cytometry plots shown are from a naïve mouse? I am surprised by how many CD45hi cells are present.

8. Supplemental Figure 4: How is it that points are being measured below the limit of detection?

9. Authors interchange “Supplemental” and “Supplementary” in both the main text and the Legends.

Reviewer #2: 1. The authors seem to argue that microglial STAT1 deletion is primarily influencing disease by preventing the activities of IFN-g signaling, though of course the actions of other interferons would also be impacted. The authors should discuss this possibility and its potential relevance to their findings, if any.

2. The finding that T cell responses are "potentiated" in microglial-stat1 knockouts seems likely to be a indirect effect of increased parasite burden, rather than a specific consequence of microglial stat1 deletion. This should be discussed (unless I've missed it).

3. Related to above - were antigen-specific T cell responses assessed? Can the authors use a tetramer staining approach to more robustly assess the T gondii-specific T cell response, to distinguish it from an influx of non-specific inflammatory cells. For the measurements of IFN-g in T cells, were cells restimulated ex-vivo using a T. gondii peptide? Some clarification of this issue is warranted. Data would be welcome, but not necessarily required.

PLOS authors have the option to publish the peer review history of their article (what does this mean?). If published, this will include your full peer review and any attached files.

Reviewer #1: No

Reviewer #2: No

Figure Files:

Data Requirements:

Reproducibility:

References:

---

## [Editor Report · Decision Letter 1]

11 Aug 2022

Dear Dr. Harris,

We are pleased to inform you that your manuscript 'Microglial STAT1-sufficiency is required for resistance to toxoplasmic encephalitis' has been provisionally accepted for publication in PLOS Pathogens.

Best regards,

Meera Goh Nair

Associate Editor

PLOS Pathogens

Kami Kim

Section Editor

PLOS Pathogens

Kasturi Haldar

Editor-in-Chief

PLOS Pathogens

orcid.org/0000-0001-5065-158X

Michael Malim

Editor-in-Chief

PLOS Pathogens

orcid.org/0000-0002-7699-2064

All comments have been addressed.
---

## [Editor Report · Acceptance letter]

24 Aug 2022

Dear Dr. Harris,

We are delighted to inform you that your manuscript, "Microglial STAT1-sufficiency is required for resistance to toxoplasmic encephalitis," has been formally accepted for publication in PLOS Pathogens.

Best regards,

Kasturi Haldar

Editor-in-Chief

PLOS Pathogens

orcid.org/0000-0001-5065-158X

Michael Malim

Editor-in-Chief

PLOS Pathogens

orcid.org/0000-0002-7699-2064